



# Wind-induced seismic noise at the Princess Elisabeth Antarctica Station

Baptiste FRANKINET [1,2], Thomas LECOCQ [1], Thierry CAMELBEECK [1]

[1] Royal Observatory of Belgium, Brussels, Belgium
[2] Glaciology Laboratory, Université Libre de Bruxelles, Brussels, Belgium

*Correspondance to Baptiste Frankinet (Baptiste.Frankinet@gmail.com)*

**Abstract.** Icequakes are the result of processes occurring within the ice mass, or between the ice and its environment. Studying icequakes provide a unique view on the ice dynamics, specifically on the basal conditions. Changes in conditions due to environmental, or climate, changes, are reflected in icequakes. Counting and characterizing icequakes is thus essential to monitor them. Most of the icequakes recorded by the seismic station at the Belgian Princess Elisabeth Antarctica

Station (PE) have small amplitudes corresponding to maximal displacements of a few nanometres. Their detection threshold is highly variable because of the rapid and strong changes in the local seismic noise level. In this study, we evaluated the influence of katabatic winds on the noise measured by the well-protected PE surface seismometer. Our purpose is to identify whether the lack of icequakes detection during some periods could be associated with variations in the processes generating them or simply to a stronger seismic noise linked to stronger wind conditions. We observed that

the wind mainly influences seismic noise at frequencies greater than 1 Hz. The seismic noise level well correlates linearly with the wind velocity, but this correlation follows different linear laws at wind velocity lower and greater than 6 m/s, with a respective variation of 0.4 dB/(m/s) and 1.4 dB/(m/s). These results allowed presenting a model and synthetic spectrogram that explain the behaviour of the wind-induced seismic noise at PE. This model enables us partially removing the influence of wind impact from the original seismic dataset, which improves the observation of cryoseismic activity near the PE

station.

## 1 Introduction: Icequakes

The study of icequakes provides insights into the different processes linked to ice dynamics. Icequakes, or cryoseisms, originate from the formation of crevasses, basal sliding, hydrofracturing, iceberg calving, in-glacier fracturing, and glacial-seismicity triggered by an earthquake. A synthesis of the main types of icequakes and their causes is presented

by Podolskiy & Walter (2016). Cryoseismic sources can have seismic signatures difficult to distinguish one from another. For example, the crevasse formation represents very short events (< 1 s) over a large frequency band (10 - 50 Hz). Crevasse formation events have a propagation velocity of 0.01 m/s up to 30 m/s and generally do not exceed 10 μm in amplitude, that make them close in amplitude to microseismic noise or to wind-induced ground motion that can cause similar seismic signature and amplitude (Bormann & Wielandt, 2013; Naderyan et al., 2016; Withers et al., 1996). Cryoseismology has

not been studied thoroughly in all regions of Antarctica but thanks to the improvement of instrumentation and the increasing number of seismic stations in Antarctica, numerous studies linking seismology to glaciology have been published in the last decade. From linking the microseismicity induced by tides in the grounding line of East-Antarctica (Barruol et al., 2013), focusing on tremors from stick-slip motions in the Whillans ice shelf (Paul Winberry et al., 2013), studying specific cryoseismic events observed at Ekström Ice Shelf, Antarctica (Hammer et al., 2015), to observing thermal icequakes and

their origins on blue ice in East-Antarctica (Lombardi et al., 2019).



## 1.1 The Antarctica Princess Elisabeth Belgian station and seismic stations

The Belgian Princess Elisabeth Antarctica Station (PE) was built during the first International Polar Year 2007-2008 and completed in 2008-2009. It is situated 300 m North of Usteinen nunatak (71°57' S, 23°20' E) on a small flat granite ridge, a few kilometers north of the Sør Rondane mountain range. The Usteinen nunatak is approximately 700 m long and 20-30 m wide and is composed of massive coarse-grained granite with minor xenolithic blocks of metamorphic rocks (Kojima & Shiraishi, 1986). To the south, the Sør Rondane Mountains peaks have an elevation up to 4000 m and form part of the Eastern Antarctica Precambrian shield (Pattyn et al., 1992). It allowed studying from meteorites to microbiology (Peeters et al., 2011; Pushkareva et al., 2018), glaciology (Callens et al., 2015; Pattyn et al., 2010), and meteorology (clouds, aerosols, temperature) (Gossart et al., 2019; Gossart et al., 2019; Herenz et al., 2019; Souverijns et al., 2018). In parallel, the Royal Observatory of Belgium installed a permanent broadband seismic station (BE.ELIB) on the bedrock near the base in February 2010 (Camelbeeck et al., 2019). This station improves the sparse coverage of seismic stations in Antarctica. Indeed, the closest seismometer is located at the Russian Novolazarevskaya base, 430 km West from PE. To the East of PE, the closest station is the Japanese Syowa site (680 km away). Because of its location, the station gave a new source of information for global seismic studies as well as for inferring the crustal structure beneath it (Camelbeeck et al., 2019). The addition of a temporary seismic network during the 2014 austral summer (see Table 1 and Figure 1) has highlighted seismic activity (Camelbeeck et al., 2019), within a radius of 150 km around the station, hitherto unknown. This seismic activity is related to the interaction between ice and bedrock or from within the ice.

## 1.2 Icequakes and seismic noise

Most icequakes can only be detected with seismometers while the resulting damage and movements of the ice sheet and associated glaciers can be observed by other geophysical or geodetic means, such as GPS (Capra et al., 1998) or radar interferometry (Mohr et al., 1998; Rignot et al., 2011). Cryoseismic catalogues and seismic observations can be correlated with numerical models of eastern Antarctic ice dynamics to quantify whether areas of specific activity are related to typical subglacial conditions (Lipovsky & Dunham, 2015; Nanni et al., 2020; F. Pattyn, 2010; Smith, 1997, 2006). These icequakes mostly have a very low seismic amplitude (few nanometres of displacement) but can still be detected owing to the very low seismic noise observed in Antarctica. Icequakes' signal-to-noise ratios (SNR) decrease when the noise increases, hence it is important to identify the noise sources and their power to evaluate the catalogue completeness before concluding the ice dynamics.

At PE, there exist a few anthropogenic noise sources like wind turbines, activity in the buildings, and human activities during the summer). The region is also subjected to rough meteorological conditions, composed of katabatic winds with sometimes velocities higher than 25 m/s (Pattyn et al., 2010). Such high-velocity winds have been known to affect the seismic data recorded (Johnson et al., 2019; Lott et al., 2017) because similarly to how tectonic earthquakes are registered by seismometers, the kinetic energy from the wind is converted into mechanical energy when it reaches the instrument enclosure which induces noise on the seismic record (Walker & Hedlin, 2010). This wind-induced seismic noise depends on wind velocity (Johnson et al., 2019). Understanding the effect of wind-induced seismic noise is crucial in monitoring icequakes as well as to understand missing icequakes in the data.

The PE and its permanent seismometer (ELIB) are relatively well protected by the mountain range from the fastest katabatic winds coming from the Antarctic Plateau. ELIB is located on the same flat granite ridge as the Princess Elisabeth Station and inside a shelter, 350 m from the base. Compared to ELIB, the temporary seismometers installed in 2014 (Figure



1) are less protected, and more prone to wind noise. If ELIB sees an increase of seismic amplitude related to wind, the
temporary seismic stations should therefore have an increased wind-induced ground motion. The base is powered by solar
panels and nine Proven Energy 6kW wind turbines (WT) (Belspo, 2007). Each consists of a 9 m high tower with a 3-blade
rotor that adapts the angle of the blades in relation to the wind speed to generate the maximum amount of power from low-
velocity winds and reduce the amount from high-speed winds. When the wind speed is low, the angle of the three blades
is reduced up to 5° and when the wind speed is the highest, the angle can increase up to 45° which reduces by half the 5.5
m rotor diameter and the resulting rotational speed. The effect of wind turbines on seismic records has also been studied in
the past and often results in noise increase in discrete frequency bands related to their shape, structure, height, the number
of blades, and rotational speed (Mucciarelli et al., 2005; Stammler & Ceranna, 2016; Withers et al., 1996). Wind-induced
seismic noise has energy at a wide range of frequencies (1 - 60 Hz and below 0.05 Hz), and its amplitude decreases rapidly
with depth (Withers et al., 1996). Wind-induced seismic noise characteristic frequencies and amplitudes also depend on
wind interaction with man-made constructions (Hillers et al., 2015; Johnson et al., 2019; McNamara, 2004; Stammler &
Ceranna, 2016). In Antarctica, with the lack of trees, the seismic noise induced by the wind should likely originates from
the interaction with the base's buildings, wind-turbines, and topography. At longer periods, the atmospheric pressure-field
can induce tilting (De Angelis & Bodin, 2012).

In this paper, we present an analysis of the influence of the wind velocity onto the seismic data from the ELIS
seismometer at the Princess Elisabeth Station. By aggregating seismic data for different wind speeds, we quantify the
relationship between wind energy and seismic ground motions. We present a model of the noise baseline when there is no
wind and its increase for each increment of wind speed, in all frequency bands. Using this model, we compute a model of
the wind-induced seismic noise for ELIB. We applied a similar model to each station of the temporary seismic network
(ANT). Finally, we used these models to evaluate the impact of the wind-noise on the detectability of icequakes

## 2 Data & Method

Our dataset includes seismic and wind velocity measurements at the PE base and seismic signals recorded between
January and April 2014 by five temporary seismic stations, the ANT network, installed in the Sør Rondane Mountains
(Figure 1).

The seismic data at the PE comes from the broadband seismic station (ELIB) installed in February 2012
(Camelbeeck et al., 2019; Lombardi et al., 2019). This station worked irregularly up to the end of 2016 due to difficulties
providing continuous power supply during the austral winter, but recordings are continuous for the years 2017, 2018, and
partly 2019 and 2020. The data collected by the ANT stations concern the period from January – April 2014. ELIB as well
as the other temporary stations except for ANT4 use Trillium 120P, 120 s seismometers that sample at 100 samples/s,
giving a recording bandwidth from 0.008 to 50 Hz, allowing to record small local seismic events as well as the worldwide
large earthquakes (Camelbeeck et al., 2019). ANT4 is a Streckeisen STS-2 gen3 120 s seismometer (owned by ETH-
Zurich, see Table 1).

The wind data comes from an Automated Weather Station (AWS) designed by the Institute for Marine and
Atmospheric Research, Utrecht University (UU/IMAU) (van den Broeke, 2006) is provided by the AEROCLOUD project.
The AWS is installed 300 m from the Princess Elisabeth Station, close to the ELIS seismometer site (see Figure 1 and
Table 2). It has been working since February 2009 and was replaced by a new AWS in December 2015, which is still in
operation. The AWS is designed to work for long periods without being serviced and offers the opportunity to measure



meteorological variables in remote areas and harsh weather conditions. These stations register wind speed, direction, temperature, humidity, atmospheric pressure at 2.0 m above the ground surface, and averaged over an hour window.

We use the seismic data from the ELIS station for the entire 2017 year (01 January 2017 - 31 December 2017), to extract hourly Power Spectral Density (PSD), which describes the power present in the signal as a function of frequency. Probabilistic PSD is then used to determine the baseline noise level (McNamara, 2004). PSDs are computed using the Obspy package (Beyreuther et al., 2010) based on the McNamara method (McNamara, 2004) which computes the PSD via a finite-range Fast Fourier Transform (FFT) of the original data (see parameters in Appendices). The PSDs are corrected for the instrument response to acceleration and converted to decibels [dB $(m/s^2)$ $^2$/Hz] to allow the comparison with the

Peterson models (1993). We apply the same processing to compute the hourly PSDs for the five stations of the ANT network.

   To quantify the effect of wind velocity onto the seismic data from ELIB, we aggregate the hourly PSDs for wind speeds ranging from 0.1 to 25 m/s. This provides a tool to generate synthetic PSDs for each wind speed, suppressing the wind-induced noise from the observed data.

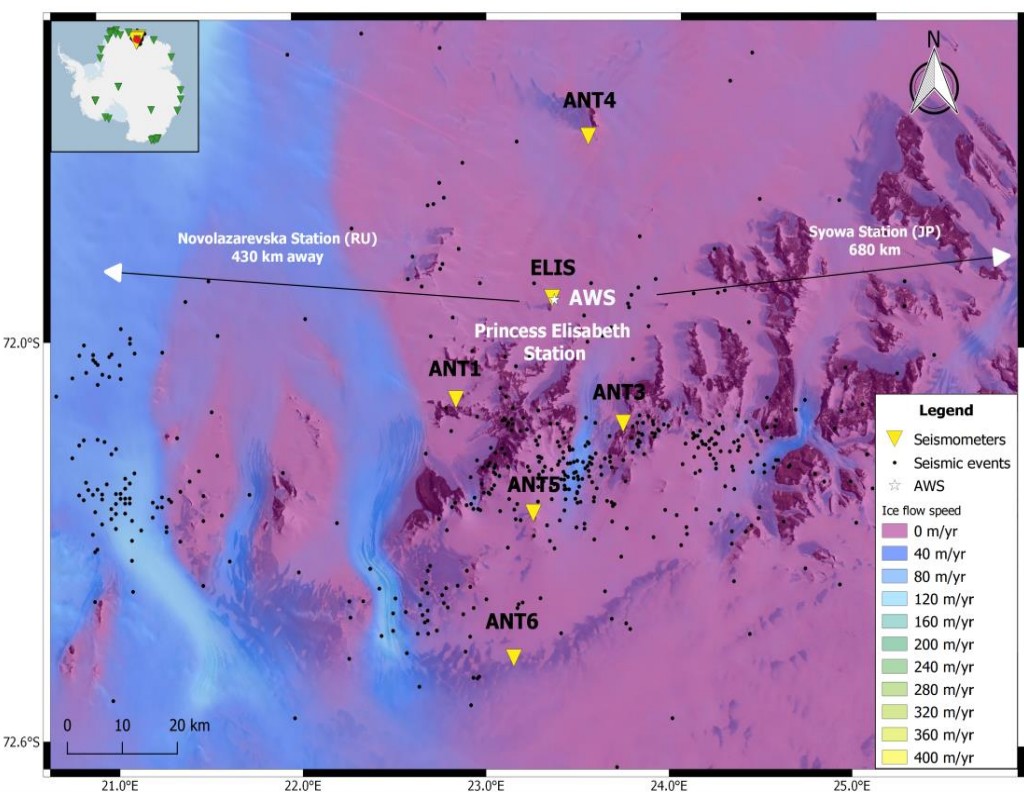

**Figure 1 : Network of instruments used in this study: 1 permanent (ELIS) and 5 temporary (ANT-) seismic stations. Other seismic stations in Antarctica reported by the International Seismograph Station Registry (http://www.isc.ac.uk/registries) are shown on the context map (green triangles) and the 2 closest from Princess Elisabeth Station are pointed towards with a white arrow: Novo. (RU) and Syowa (JP) (map was drawn with QGIS, using the Quantarctica module (Matsuoka et al., 2013). The**
**cryoseismic activity (Camelbeeck et al., 2019) registered by a minimum of 4 stations during the working time of the ANT network (January - May 2014) is shown by the small dark circles (Ice flow speed from Mouginot et al., 2019).**





| Station | Instrument | Location | Latitude [°] | Longitude [°] | Elevation [m] | Start | End |
|---|---|---|---|---|---|---|---|
| ELIB | Nanometrics Trillium 120P, 120 s | Princess Elisabeth Antarctica Station (borehole) | -71.947 | 23.346 | 1359 | 2010-02-15 | 2014-06-13 |
| ELIS | Trillium 120P, 120 s | Princess Elisabeth Antarctica Station | -71.947 | 23.347 | 1372 | 2012-02-11 | In service |
| ANT1 | Trillium 120P, 120 s | Otto | -72.099 | 22.840 | 1718 | 2014-01-02 | 2014-04-14 |
| ANT3 | Trillium 120P, 120 s | Gunnestadbreen (outlet glacier) | -72.134 | 23.727 | 1397 | 2014-01-04 | 2014-04-14 |
| ANT4 | Streckeisen STS-2 gen3 | , Vesthaugen hill (west hill) | -71.703 | 23.529 | 1217 | 2014-01-25 | 2014-08-25 |
| ANT5 | Trillium 120P, 120 s | Last Nunatak | -72.271 | 23.252 | 2366 | 2014-01-07 | 2014-03-31 |
| ANT6 | Trillium 120P, 120 s | Blue-Ice | -72.488 | 23.150 | 2379 | 2014-01-07 | 2014-12-05 |

**Table 1: Belgian Antarctica Seismometers information.**

| Station | Instrument | Range +- accuracy | Direction +- Accuracy | Latitude | Longitude | Start | End |
|---|---|---|---|---|---|---|---|
| AWS | Young 05103 | 0 to 60 m/s +- 0.3 m/s | 0 to 360° +- 3° | -71° 94' 9" | 23° 35' 8" | 2009-02-02 | In service |

**Table 2: Automatic Weather Station (AWS) information from the AEROCLOUD Project.**

### 3 Wind Induced Noise Model for ELIS

To quantify the link between wind velocity and seismic noise at the PE base, we computed hourly PSDs of the ELIB vertical seismometer for the whole year 2017 and extracted the 5th percentile amplitude for every 0.25 m/s wind speeds between 0 and 25 m/s (Figure 2). The wind speed used in this study is the maximum average wind speed recorded by the AWS which is measured at 2 m height every 10 minutes and averaged for each hour of 2017. The 5th percentile is preferred over the average to define base noise levels for each wind speed step without taking outliers into account. The wind speed steps and their base noise amplitude exhibit (Figure 2) an increase of noise amplitude at all periods but the effect is stronger below 2 s and above 10 s.

The seismic noise levels increase with the wind velocity and exhibit two different behaviour for wind velocity greater and smaller than 6 m/s. The increase of seismic noise is moderate for wind velocity from 0 to 6 m/s and larger above 6 m/s. At 0.1 s (10 Hz) there is a 42 dB difference between 0 and 25 m/s, which corresponds to a ground velocity increase of 100 times. The wind-noise effect is higher on the horizontal components than on the vertical component of the seismometer. This has been already observed and is due to the direct interaction of the wind travelling direction (horizontal) to the seismometer inducing tilt noise (Mucciarelli et al., 2005).

To create the synthetic noise model, we need to quantify seismic noise changes at each frequency with respect to the wind speed amplitude. For each period band, two linear relationships are determined between 0 and 6 m/s and above 6 m/s (slopes: $a_l1$ and $a_l2$ in Figure 3a). The data used for the weighted regression is the 5th percentile of wind speeds binned by 0.25 m/s with a minimum of 10 observations per bin. The weights are defined as the inverse of the standard deviation within each bin. For example, Figure 3b shows the two linear regressions at the 0.26 s period (dashed vertical line on Figure 3a): the wind-induced noise increases by approximately 2dB from 0 to 6 m/s, and after 6 dB it increases by 1.5 dB/m/s.



The lower number of occurrences of wind speeds above 10 m/s could lead to instability, but the regression drawn between 6 and 10 m/s is robust and fits the observations at higher wind speed. The linear regressions are computed for every frequency and therefore describe the behaviour of the seismic noise induced by the wind at ELIS.

Once the linear parameters are determined for each period of the spectrum, we can run the model for any theoretical wind speed to obtain a synthetic PSD spectrum. Doing this for different wind speeds, we generate a synthetic spectrogram (Figure
4a). Transforming the synthetic PSDs to seismic velocity amplitude requires integration to PSDs of velocity and the application of Parseval's theorem that links the power spectrum and the RMS (root mean square) of a signal. The RMS velocity calculated in the 1-50 Hz frequency band (Figure 4b), i.e. the band where most cryoseismicity is expected to occur, shows an exponential increase from 0.2 to 2.8 µm/s between 0 and 25 m/s wind speed. Figure 4b also shows the frequency band (8-50 Hz) and the amplitudes (smaller than 0.3 µm) of the icequakes signals studied by Lombardi et al. (2019). This
illustrates that, based on our model, Lombardi et al. (2019) are vulnerable to missing seismic event when the wind speed exceeded 10 m/s.

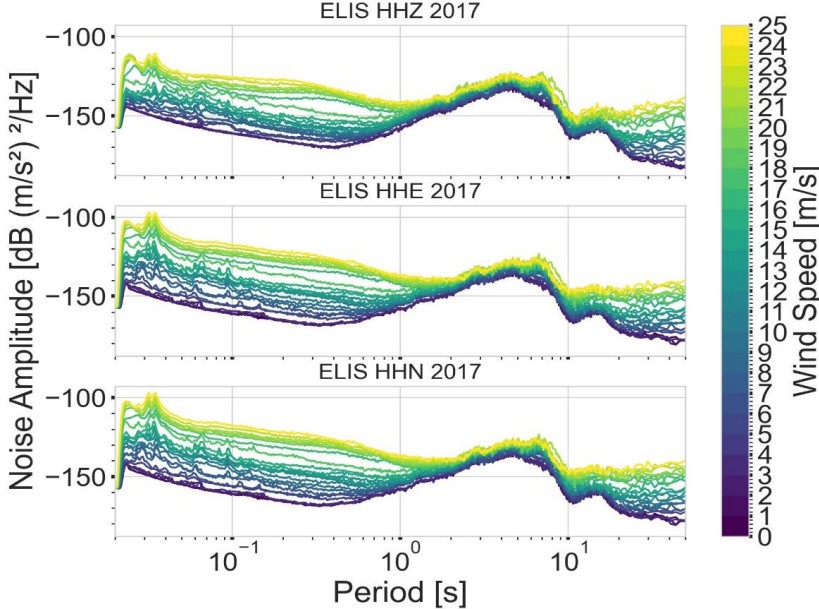

**Figure 2: 2017 ELIS PSDs computed for 0 – 25 m/s wind speed. HHZ, HHE and HHN are seismometers codename; 1ˢᵗ letter represents the sampling rate and response band of the instrument: H = High-Broadband seismometer; 2ⁿᵈ letter is the family of
170 the sensor: H = High-gain seismometer; 3ʳᵈ letter Z, E and N represent the vertical, East-West and North-South motion respectively.**





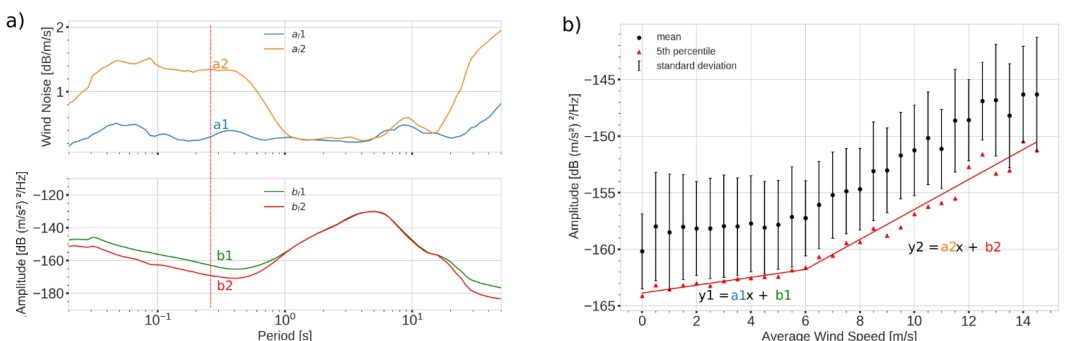

**Figure 3: a): Wind Induced Noise Model describing the parameters of 2 linear functions for every period of ELIS HHZ in 2017: y = ax + b: The first subplot represents the "a" parameter and the second one the b parameter. Those 2 different linear relation parameters are until 6 m/s (a₁1/b₁1) and after 6 m/s (a₁2/b₁2). B): Behaviour of the 5ᵗʰ percentile noise amplitude for the 0.26 s period at ELIS HHZ for the average wind speed of 2017. There are 2 different linear behaviours: before 6 m/s and after 6 m/s respectively y1 and y2: the parameters used are shown on the graph a) by the red line and their parameters.**

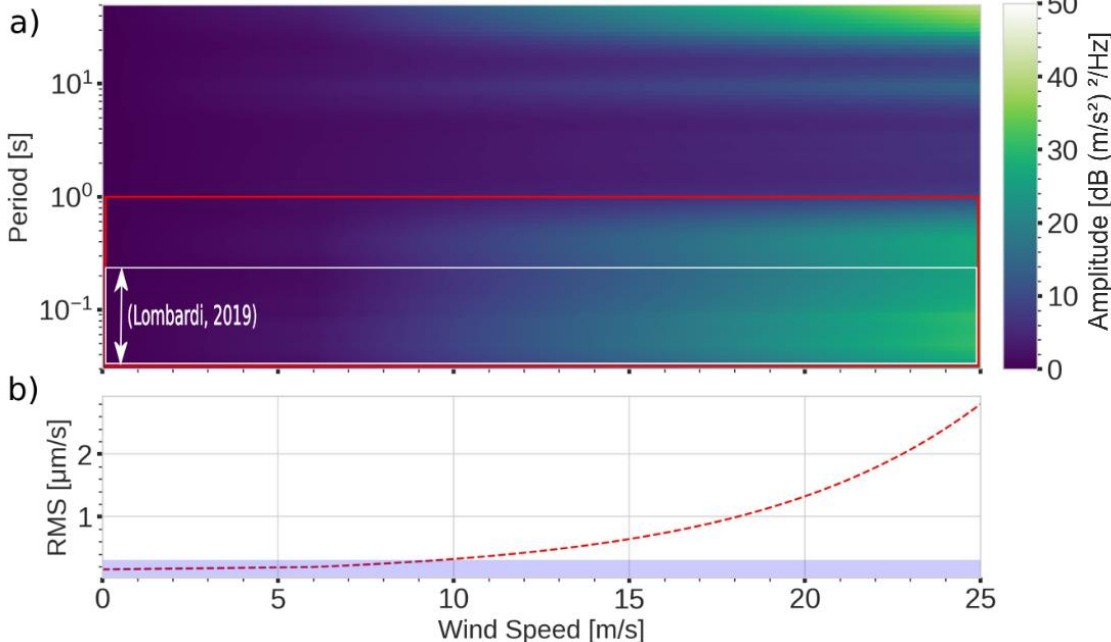

**Figure 4: a) Relative Synthetic Spectrogram representing wind-induced noise increase; in blue is the frequency range of the thermal icequakes observed by Lombardi et al., 2019 and b) the 1 – 50 Hz RMS ground speed at ELIS HHZ extracted from the red rectangle on subplot a) and in light blue the amplitude range of these icequakes.**



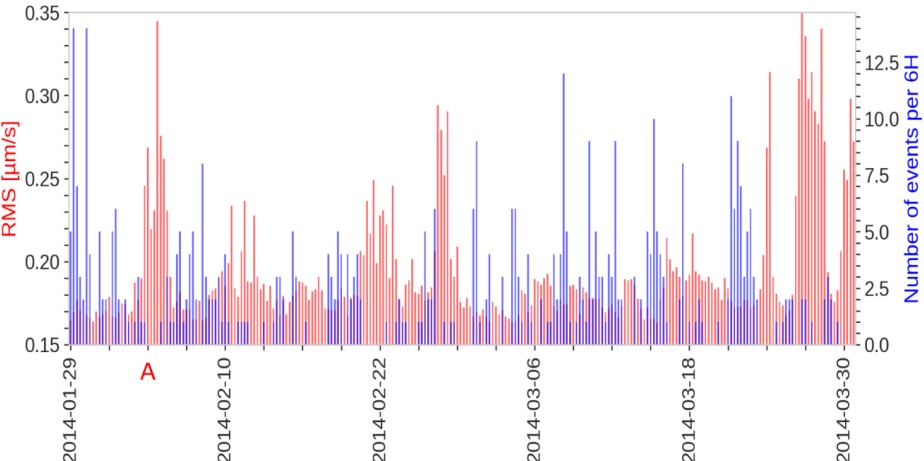

**Figure 5: RMS per 6h calculated from the average wind speed and the noise model seen at ELIS vs the cumulative number of events detected per 6h (seismic rate) by Camelbeeck et al. (2019)**

### 3.1 Seismic Noise for the ANT network

We used our model to evaluate the completeness of the catalogue of icequakes identified between January and April 2014 by the ANT temporary seismic network, including ELIB station. The cumulative number of seismic events detected per 6 hours located by Camelbeeck et al. (2019), shows an inverse correlation with the seismic noise level deduced by our model (RMS) from the wind speed measured at PE (Figure 5). This RMS is calculated from the mean wind speed registered by the AWS averaged per 6h using our model shown in Figure 4b. This inverse correlation suggests that the variation in the icequakes activity rate would be directly related to seismic noise conditions induced by the wind. For example, from the 3rd to the 4th of February 2014 (A), the RMS increase from 0.17 to 0.35 µm/s at the maximum peak. Over the period presented in Figure 5 (29th January 2014 – 30th March 2014), 472 events were manually detected by Thierry Camelbeeck. If these events were equally distributed, 7.9 events could be detected each day but, the numbers of events found over this period were not, as during the 3rd February: 5 events were detected and, on the 4th, when the RMS increased drastically, no events were recorded. We can therefore observe that a quiet station improves the detectability of icequakes.

ELIB is located in an area where the ice sheet moves very slowly, which is not the case for some of the temporary stations of the ANT network. Therefore, the ice sheet movements did not contribute greatly to the recorded seismic noise level at the ELIB seismic station. Using the 2014 data from the AWS, we can compute a synthetic seismogram for the wind-generated noise during the deployment of the temporary ANT network. Assuming that the wind field and its effects on ELIS are identical at the other seismometers, which might not necessarily hold true, we obtain "clean" spectrograms by withdrawing the frequency-dependent noise increase due to wind. Figure 6a-f shows the wind-corrected spectrograms for the 6 stations, which should highlight the contribution of the cryoseismic activity of the East-Antarctic ice-sheet in the seismic noise of each station. Figure 6g-l represents the RMS velocity of the 6 previous spectrograms. For comparison, the average wind speed and temperature recorded at ELIS (Figure 6m-n) are shown.



The periods where the RMS velocity of the stations shows a significant cryoseismic activity are indicated by blue arrows labelled A1 to A7 (Figure 6l). Some stations, and particularly ANT3 and ANT6, still exhibit a correlation between the increase of wind speed and their remaining RMS velocity.

Between the 8 - 20 January (A1), there is a small co-increase of wind speed and RMS velocity, especially at ANT3. The same thing happens during the 8 - 15 February (A2) and 20 – 26 February (A3) intervals. 1 – 7 March (A4) shows RMS velocity peaks for all stations but ELIS, ANT1, and ANT4. The 13 – 15 March (A5) starts with a strong activity at ANT3 and then an increase in the other stations, including strong peaks at ANT6. The 18 - 20 March period (A6) as more energy on ANT3 than the other stations. The 23 – 27 February period (A7) is again dominated by strong energy at ANT3 but other stations peak during that period too (ANT6 and ANT1 particularly). The activity of ANT3 seems, in general, to be higher than on the other stations, between 1 – 28 February and after 15 March (end of A5) until the rest of the deployment.

In addition to the activity spanning over several days, the six stations show a strong diurnal activity which has been shown to be linked to temperature variation that induces thermal icequakes (Lombardi et al., 2019). Its intensity is larger at ANT6 throughout the deployment period. At all stations but especially at ELIS and ANT1, the diurnal effect seems to lessen after 8 March 2014 (A4).

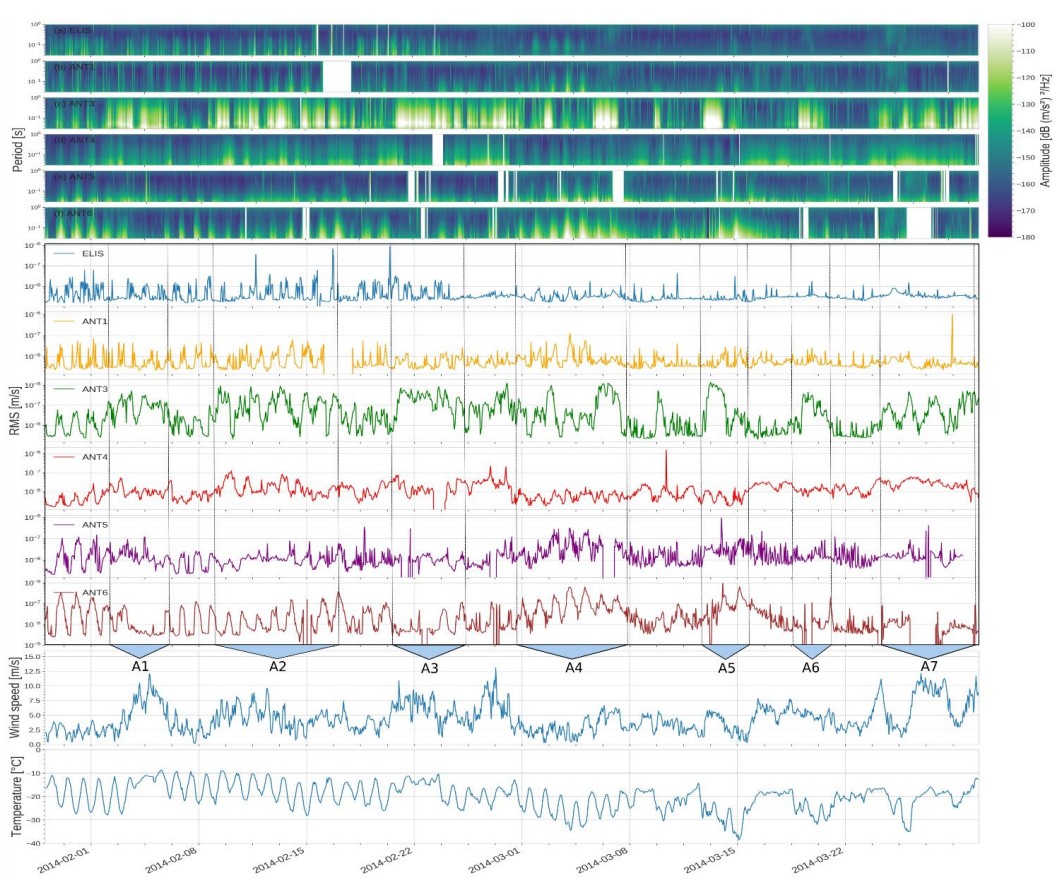

**Figure 6: Belgian Antarctica Network Spectrograms (1 – 50 Hz) without wind noise for the 29/01/2014 – 30/03/2014 period with their associated RMS and the average wind speed for the same period from the AWS station at ELIS.**

## 4 Discussion

The origin of the diurnal activity could come from thermal icequakes resulting from diurnal temperature difference as observed and studied at ELIS (Lombardi et al., 2019). On all stations but especially at ANT5 and ANT6, there is a greater diurnal activity during A4 and A5, which correlates from a sudden drop of temperature and a greater temperature difference between the daytime/night-time (Figure 6). This also suggests that part of the greater activity seen at ANT3 and ANT4 is most likely caused by a greater cryoseismic activity induced by the temperature change between the daytime/night-time. During A6 and A7, the same effect is observed, the activity greatly increased at ANT3 together with temperature deltas of about 10°C. The stronger activity diurnal activity at ANT6 can be explained by its setting: it is placed on blue ice and is, therefore, better coupled to register crevassing and thermal icequakes than the other stations on rock (Trnkoczy et al., 2012). The frequency content of this diurnal activity on ANT5 is higher frequency than on the other stations, most of the energy release is above 30 Hz.

The ANT3 station has a much higher amplitude of the seismic noise than any other station from the network. The activity at ANT3 seems to correlate with the wind for at least A1, A2, A3, and A7 periods. This indicates that to a certain extent the wind field at those stations is the same as at ELIS but that the wind strength and/or its effect on the seismic noise is





greater. Nevertheless, certain peaks have a high amplitude that seems hard to link to the wind activity, at least not the same wind like the one measured in ELIS. For example, to reach the peaks at 1.0 μm/s seen at ANT3, in A2, A3, and A4, the wind needs to reach a speed at least 17.5 m/s, which was not observed at ELIS during the period where the ANT network was deployed. The maximum wind speed during that period was 14 m/s. Another cause of the difference in energy could be linked to the insulation or coupling difference of the seismometer in the different stations. Part of the explanation for site-specific winds can be explained by the location of ANT3 close to an outlet glacier, which could channelize the winds originating from the Plateau to the south.

The continuously higher energy at ANT4 follows the same general long-term trends as the wind speed. There seems therefore to be a small under-correction of the wind effect on the noise, caused either by slightly stronger local winds, or a slightly steeper relationship between wind and noise caused by coupling or installation settings. Stronger continuous cryoseismic activity could also explain the observations but based on Lombardi et al. (2019) we would expect to see more diurnal variation if this activity was thermally induced.

From the stations in the network, ANT1, ANT3, and ANT5 are the closest to the a priori more seismogenic zones: the collision between the glaciers and the mountains and the zone of channelized glaciers with greater ice flow speed (Figure 1).

## 5 Conclusions

Near the Princess Elisabeth Station, we observe a wind-induced seismic noise that in some cases surpasses the detectability of most icequakes. The detection of icequakes can be altered by wind speed as low as 5 m/s, as they will be hidden in the wind-induced noise. When these winds reach their highest speeds, of up to 25 m/s, the seismometer registers an increase of 100 times the seismic velocity of a stand-still moment (0.5 μm/s to 3 μm/s) making most of the small icequakes undetectable. Understanding the effect of wind-induced seismic noise is therefore crucial in monitoring icequakes as well as to understand missing icequakes in the data. To mitigate wind-induced noise and improve the quality and detectability of icequakes, we suggest, whenever possible, preferentially installing future seismometers into boreholes, far from structures that could be affected by wind, in a wind protected area. In all case, we recommend installing a meteorological station next to each instrument site to obtain local measurements of the fields.

Using the data from the permanent seismic station ELIS, we provide a synthetic model that simulates the ground motion spectrum for different wind speeds. For half of the period during which the temporary ANT network was deployed, the ANT3 seismometer exhibits greater amplitude than the other stations which can only be partially explained by greater local wind speeds. After correcting the seismograms for wind-induced noise and computing the RMS velocity, we found that some peaks still follow the wind speed in some cases but most of the time it stays independent from it.

As observed elsewhere, we suggest that the diurnal changes of energy observed are linked to cycles of cryoseismic activity induced by the large diurnal temperature delta between the daytime/night-time. The longer-lasting energy releases, on the other hand, could have different causes related or not to wind. They could either originate from slightly different wind fields, wind speeds, or couplings; or from an increased cryoseismic activity occurring in the vicinity of the station, independently from the diurnal and thermal effects, for example, crevasses or basal stick-slip events. The diurnal seismic energy at a higher frequency on ANT5 could result from different source mechanisms, with smaller, shorter icequakes occurring in the direct vicinity of the station. This could be confirmed by studying and comparing the icequakes signatures on the different stations in future work.





*Appendices*

PSD parameters: we binned PSDs for every hour-segment of the entire year. Parameters such as a 3600 s time window length were used for the PSD computation, which shows that long time seismic series are required to compute significant

PSD plots and also correspond to the wind data of the AWS station we will compare it to. We did not overlap the time series to have exact PSDs estimates for each hour. Other parameters, such as "period_smoothing_width_octaves = 0.025" means that the PSD is averaged over 1/40 of an octave at each central frequency/period. We limited our computations to 50 Hz which is known as the Nyquist frequency of the seismometers. Those parameters are important to identify noise characteristics buried in the noise such as weak seismic peaks to differentiate each PSD with more details.

Power Spectral Density parameters calculations:

    tr.stats, metadata=sp,

    ppsd_length=3600, overlap=0.0,

    period_smoothing_width_octaves=0.025,

    period_step_octaves=0.0125,

period_limits=(0.02, 50),

    db_bins=(-200, 20, 0.5)

*Acknowledgments*

We acknowledge the AEROCLOUD project (https://ees.kuleuven.be/hydrant/aerocloud/) for providing the wind data from

295 the Automated Weather Station (AWS) located at the Princess Elisabeth Station in Antarctica. The seismic data from the permanent ELIS station as well as from the ANT array should be made available via the ORFEUS data center soon, but in the meantime, access can be given directly from the ROB FDSN web services. Baptiste Frankinet acknowledges the financial support from the ROB.



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
