# Peer review of "Wind-induced seismic noise at the Princess Elisabeth Antarctica Station"

_The Cryosphere, 2020_

## Referee Comment (RC1) · Anonymous Referee #1 · 16 Dec 2020

In their study, Frankinet et al. investigate the impact of wind on the seismic noise levels measured at the Princess Elisabeth Station in Antarctica. They find that the seismic power increases with the wind speed and they determine empirical linear scaling laws between the two quantities that seem to be different for low wind speeds (<6m/s) and high (>6m/s) wind speeds. With these scaling laws, they develop a noise model as a function of frequency and wind speed. This model suggests, that icequake detection rates from a previous study at Princess Elisabeth station are biased by variable wind noise levels masking events at high wind speeds. The authors use the noise model constructed from a single station to correct its spectrograms and those from five other stations in the same region for wind noise, and discuss the resulting RMS amplitudes in the light of cryoseismic activity.

[Figure]

Because icequakes and other cryoseismic signals are often characterized by low amplitudes, detection limits and temporal variations thereof are crucial for the conclusions drawn. In this regard, the study by Frankinet et al presents a valuable contribution as it quantifies the changing noise level due to wind potentially masking events of interest. The derived wind-noise model appears reasonable and it may help future studies to evaluate the influence of wind on seismic measurements, even though its wider applicability remains unclear, as the model is derived from a single seismic station. In some parts, the manuscript needs to be substantially improved in terms of explaining/clarifying the analysis and discussing the results as the calculation of the wind-noise model and the robustness of the results remain unclear. I feel that seismic power and seismic activity (in terms of events) is mixed up in the discussion and I think that the contribution of the wind-corrected spectrograms is very limited as they hardly give insights into the glaciological processes. I therefore have several major comments and a range of smaller issues, which should be addressed before consideration for publication.

MAJOR COMMENTS

L56-58: I think that this sentence does not correctly summarize the cited studies. Seismic observations help to constrain subglacial properties, but it is to date not possible to model/link seismicity with ice flow modeling. Also, the study of Nanni et al. investigates an Alpine glacier, not Antarctica. Please correct this.

L143 and the following: In this part of the manuscript, the basis of the wind-noise model is formulated, i.e. that seismic power scales with wind velocity at two different relations for find velocity greater and smaller than 6 m/s. However, this needs to be better supported with data, as this is only shown for a single frequency bin. I suggest to plot the 5th percentile (or median) measurements (red triangles in Fig. 3b) for for the whole frequency band (color-coded). This should give further evidence of the wind-induced noise as a function of frequency. By looking at Fig. 3B, one could also conclude, that

seismic power is just dependent on wind for velocities greater 5m/s. Also, by looking at Fig. 2, there seems to be only a small increase in seismic power for wind speeds greater than approximately 20m/s.

L158-160: I think this part needs to introduce the model formula, which is used to calculate the output shown in Fig. 4a. First, the formulas for the linear regressions should be connected to the measured quantities (y=Amplitude [dB], x=wind speed [m/s]). Then, the parameters determined from the regression are used to create the model, which must be something like a1(f)*x (for x<6m/s) and a1(f)*6m/s + a2(f)*x + b2(f) (for x>6m/s), I guess. These are crucial details, which need to be added to the manuscript.

L188 and following: To further stress the point that increased wind speeds result in reduced event detections, I suggest to look at the recorded data and plot the event detections as a function of wind speed. This should show a drop in detections at higher wind speeds, in case the wind doesn't affect the icequake generation processes. Also, the red bars in Fig. 5 may actually be replaced (or compared) by the measured RMS of ELIS, which I assume is not much affected by the few short duration events per 6 hours.

L199 and following: Here, the wind speed measured at the base station is used to calculate wind-induced noise power, which is subtracted also from the five stations of the temporary network. The temporal variability of the wind-corrected PSDs are then discussed. I think that such an analysis is not well justified, as wind speeds might not be well correlated at the sites (as also indicated in the manuscript). For instance, station ANT6 is separated by about 50 km from the weather station and on the other side of a 4000 m high mountain range, which I expect to clearly influence wind conditions. In addition, the authors find that wind-corrected PSDs are still correlated with wind. This is not surprising considering that the wind-noise model is calculated from the 5th percentile of PSD observations, hence removing not the full contribution. These shortcomings must be discussed in the manuscript. Currently, this is only briefly

picked up in the conclusions.

L225 and following: This section discusses the temporal PSD/RMS variations of the wind-corrected stations, but I think that this section does not have a very profound basis given the issues raised in the previous point. Also, I doubt the usefulness of analyzing RMS amplitudes in the context of discrete icequake events. The events presented in Lombardi et al. 2019 are of short duration (<1s) and weak amplitude (<1e-6m/s), hence, I do not expect them to cause a significant contribution to the RMS amplitude. Yet, this could be checked by running a simple STA/LTA trigger on e.g. station ANT6 during high RMS amplitude periods, which is argued to register more seismicity due to its deployment on blue ice. Overall, I feel that the discussion of RMS amplitudes, does not yield useful insights into glaciologically relevant processes. Given the uncertainties and potential overinterpretation, I suggest to significantly shorten the discussion of RMS variations in the light of ice flow dynamics. Instead, I suggest to discuss some other aspects as detailed in the following comment.

Wind-induced noise levels: I am missing a discussion of the wind-noise levels and comparison to other studies. For instance, how do the results compare to the findings of the cited study by Lott et al. (2017), who also analyze wind-induced noise as a function of wind speed? In this context, it would also be helpful to discuss the wider applicability of the derived noise level. Is it also applicable to other sites in Antarctica? If available, it would be also very interesting to study other colocated seismic and weather stations.

ELIB vs ELIS: Sometimes, the manuscript refers to station ELIS, sometims to ELIB. According to Table 1, these are two different stations, with ELIB being a borehole station, yet, the existence of such a station is not mentioned in the text. This issue needs to be clarified. Actually, it would be very interesting to see Figure 2 for both surface and borehole station to evaluate the effect of a shallow borehole (according to Table 1, ELIB sits in a depth of roughly 10m?) on the wind-noise level.

[Figure]

MINOR COMMENTS

L23: in-glacier > englacial

L33: Whillans ice shelf > Whillans ice stream

L33: Reference formatting wrong, remove first name of author.

L37: Maybe use PEAS for the Princess Elisabeth Antarctica Station, as referring to PE, i.e. Princess Elisabeth, in the text is a bit misleading ;-)

L38: Start the sentence with "The PE/PEAS allowed investigations in the field of ..." and maybe give a bit more context on the meteorites as it the connection between Antarctica and meteorites is not obvious.

L46: Rewrite "... improves the sparse coverage of seismic stations ...".

L51: Move the reference to the end of the sentence and consider deleting "hitherto unknown" as it is expected to encounter microseismicity in glacierized terrain.

L54-56: Be a bit more specific here and distinguish between elastic deformation and static or plastic deformation.

L63-64: Is there "activity in the buildings" also in winter, when no humans are (apparently) around? Please specify.

L66: The link to tectonic earthquakes is misleading here, please delete.

L86: originates > originate

L90: aggregating > sorting (also in the following of the manuscript)

L99: Before it was mentioned, that the station was installed in 2010?!

L102: What does partly continuous data mean? Please sepcify.

L103: samples/s > Hz

L104-105: worldwide large earthquakes > teleseismic earthquakes

L108: "and" is missing, change to ". . . and is provided . . ."

L108: Provide reference to the AEROCLOUD project.

L113: Remove "and" before "averaged".

L115: power > seismic power

L116: What do you mean by baseline noise model? Maybe just mention that the Probabilistic PSD represents a statistical distribution of the PSDs.

L118-119: You correct the ground motion time series for the instrument response and then calculate the PSDs, right? At least that's what implemented in obspy, I think.

L120: Be more specific (it is the new high/low noise level) and correct the reference formatting. In general, I think this whole paragraph (L114-L121) could be written more succinctly.

L122: onto > on

L122-124: I think these lines can be removed, as the next section already starts with the same information. Also the second sentence is a bit out of context and I guess it should be rather "removing" or something similar instead of "suppressing" the wind-induced noise from the observed data.

L137: ". . . for every 0.25 m/s wind speed ..." > for every 0.25 m/s wide bin of wind speed

L138: What do you mean by "maximum average"?

L138: What happens at wind velocities greater than 25m/s? The presented analysis suggests, that at least 10 observations at 25m/s are available, so I would guess that also observations for even higher velocities are available.

L145: As the 42dB refer to acceleration PSD, it should be a 100-fold increase in ground

acceleration not velocity, right?

L160: Please provide a more information here, how you convert the PSD to the RMS. For instance, how do you integrate the acceleration PSD?

L164: Are the given amplitudes of the Lombardi study absolute amplitudes or also RMS amplitudes? For comparison with the wind-noise amplitude in RMS this is important, please specify. Also, the unit is wrong: 0.3micrometer > 0.3 micrometer/s.

L193: "icequakes activity rate" > icequake rate

L195: "... 472 events were manually detected by Thierry Camelbeeck." Does this refer to another study?

L196-198: The meaning here is clear, but the sentence needs to be rewritten. The last sentence ("We can therefore observe ...") can be deleted, as this is a general statement not specific to this study.

L199: From Fig. 1 it actually looks like all stations are installed in slow-moving (<40m/year) areas?

L255: Delete "a". Change "... surpasses the detectability ..." to "... prevents the detection ...".

L256: Delete "most"

L258: I think there is again a problem with the units (ground velocity vs acceleration vs RMS amplitude), at least the 100-fold increase in ground velocity does not fit the "0.5micrometers/s to 3micrometers/s" statement.

L261: Delete "future".

GENERAL COMMENTS AND FIGURES

The introduction contains a lot of detail on the study site already, which could be moved

to section 2.

I suggest to use the term "ground velocity" instead of "seismic velocity", as the latter typically refers to material properties, i.e. seismic velocity in rock. Also, I suggest to use "thermally-induced icequakes" instead of "thermal icequakes".

The references contain some flaws and should be carefully checked.

Fonts of Fig. 3 and Fig. 6 are too small.

Table 1 and 2 may be merged and the coordinates given in Table 2 adjusted to the format of those in Table 1.

Fig. 2, caption: explanation of channel names is not relevant and can be removed.

Fig. 3: From a conceptual point of view, it makes more sense to switch a) and b). The caption should be rewritten to make it better understandable. Also, 3a does not yet show the noise model, right?

Fig. 4a: "(Lombardi, 2019)" → "Lombardi et al., 2019". Caption, first line: blue > white. Maybe also split in two sentences and reformulate.

Fig. 5, red y-label: change to "Modeled RMS" for clarity.

Fig. 6: labels a) – n) are missing. Also, provide more detail in the caption, e.g. explain blue arrows.

---

## Referee Comment (RC2) · Anonymous Referee #2 · 19 Dec 2020

I read the paper "Wind-induced seismic noise at the Princess Elisabeth Antarctica Station" with great interest. The authors quantified the relationship between wind energy and seismic ground motions, developing a model of the wind-induced seismic noise associated with icequakes.

My general impression is that the article is rather well written: only a specific part of the Discussion and the Appendix needs to be rephrased. As highlighted by the authors, the study of icequakes provides insights into the different processes linked to ice dynamics. The quantitative analysis of icequakes using ambient noise data processing techniques represents an important application of the studies concerning noise wavefield. Then, in my opinion the paper can be published; however, it needs a minor revision before being accepted.

Specific comments – the parts that needs to be rephrased are underlined in the highlighted manuscript. 1) Lines 87 and 88: The sentence "By aggregating seismic data for different wind speeds, we quantify the relationship between wind energy and seismic ground motions" should be supported by more references, e.g. "Lepore et al. (2016), Impact of wind on ambient noise recorded by seismic array in northern Poland, Geophys. J. Int., 205, 1406-1413"; 2) Lines 248 and 249: the sentence "the collision between the glaciers and the mountains and the zone of channelized glaciers with greater ice flow speed" is not clear. Rewrite it; 3) The Appendix is badly written and in some parts it is not clear. Rewrite it.

Technical corrections – in the highlighted manuscript, the parts that should be deleted are crossed out and marked in purple, while the parts that need to be corrected are marked in yellow. Corrections are reported in the manuscript in the shape of pop-up notes. Therefore, here the lines are listed in which the parts needing modifications are present. 1) ABSTRACT, lines 6 and 7; 2) INTRODUCTION: ICEQUAKES, line 23; 3) ICEQUAKES AND SEISMIC NOISE, lines 51, 60, 63, and 83; 4) DATA & METHOD, lines 99, 114, 116, and 118; 5) WIND INDUCED NOISE MODEL FOR ELIS, lines 139, 140, 149, and 161; 6) DISCUSSION, lines 228, 231, 246, 247 and 248; 7) CONCLUSIONS, line 269.

Please also note the supplement to this comment:
https://tc.copernicus.org/preprints/tc-2020-267/tc-2020-267-RC2-supplement.pdf

---

## Author Comment (AC1) · 14 Jan 2021

Hello,

Thank you very much for your review.

We corrected our manuscript according to your yellow and purples comments in your review.

1. I also modified lines 87 and 88 by adding 2 references that talk about the relationship between the wind energy and ground motion;

"Lepore, S., Markowicz, K. & Grad, M.: Impact of wind on ambient noise recorded by seismic array in northern Poland, Geophysical Journal International, Volume 205,

Issue 3, Pages 1406–1413, https://doi.org/10.1093/gji/ggw093, 2016"

and

"Johnson, C. W., Meng, H., Vernon, F., & Ben‐Zion, Y.: Characteristics of Ground Motion Generated by Wind Interaction With Trees, Structures, and Other Surface Obstacles. Journal of Geophysical Research: Solid Earth, 2018JB017151, https://doi.org/10.1029/2018JB017151, 2019."

2. I rewrote: "From the stations in the network, ANT1, ANT3, and ANT5 are the closest to the more seismogenic zones: the collision between the glaciers and the mountains and the zone of channelized glaciers with greater ice flow speed"

by using bullet points to make it clearer.

From the stations in the network, ANT1, ANT3, and ANT5 are the closest to the more seismogenic zones: 1) The collision zone between the glaciers and the mountains. 2) The channelized glaciers represented by greater ice flow speed (Figure 1).

3. I modified the appendix to make it simpler to understand.

"To calculate the PSD parameters used in our models, we used PSDs for every hour-segment of the entire year. As parameters, a 3600 s time window length was used for the PSD computation, which shows that long time seismic series is required to compute significant PSD plots and also correspond to the wind data of the AWS station we compared it to. We did not overlap the time series to have exact PSDs estimates for each hour. The PSD is also averaged over 1/40 of an octave at each central frequency/period. We limited our computations to 50 Hz which is known as the Nyquist frequency of the seismometers used. Those parameters are important to identify characteristics buried in the noise such as weak seismic peaks and differentiate each PSD in more details."

Thank you in advance,

Baptiste Frankinet.

---

## Author Comment (AC2) · 25 Apr 2021

Dear Reviewer 1,

Sorry for the time taken for doing this review. Thank you very much for your review, which helped us improve the manuscript. Please find answers to your questions here below.

Baptiste Frankinet, for the authors. — Major comments: ## L56-58: I think that this sentence does not correctly summarize the cited studies. Seismic observations help to constrain subglacial properties, but it is to date not possible to model/link seismicity with ice flow modeling. Also, the study of Nanni et al. investigates an Alpine glacier,

not Antarctica. Please correct this.## Thank you for this remark, indeed, the study of "Nanni et al" is not directly correlated and we removed it, and corrected the sentence: "Cryoseismic catalogues and seismic observations can be correlated with numerical models of eastern Antarctic ice dynamics to constrain subglacial properties of a specific area." ## L143 and the following: In this part of the manuscript, the basis of the wind-noise model is formulated, i.e. that seismic power scales with wind velocity at two different relations for find velocity greater and smaller than 6 m/s. However, this needs to be better supported with data, as this is only shown for a single frequency bin. I suggest to plot the 5th percentile (or median) measurements (red triangles in Fig. 3b) for for the whole frequency band (color-coded). This should give further evidence of the wind-induced noise as a function of frequency. By looking at Fig. 3B, one could also conclude, that seismic power is just dependent on wind for velocities greater 5m/s. Also, by looking at Fig. 2, there seems to be only a small increase in seismic power for wind speeds greater than approximately 20m/s.## Thank you for this comment. Our model parameters (a1, a2, b1 & b2) are computed for all frequency bins and for each 0.5 m/s step. They are represented on Figure 3a, which shows a strong frequency-dependent relation. In turn, this means that computing the simulated RMS values is important when comparing with seismic rate. On the attached Figure R1, we plot an example with showing other frequencies in the example of Figure3b, but we fear this makes the graph much less readable, and redundant with Figure 3a.

**L158-160: I think this part needs to introduce the model formula, which is used to calculate the output shown in Fig. 4a. First, the formulas for the linear regressions should be connected to the measured quantities (y=Amplitude [dB], x=wind speed [m/s]). Then, the parameters determined from the regression are used to create the model, which must be something like a1(f)*x (for x<6m/s) and a1(f)*6m/s + a2(f)*x + b2(f) (for x>6m/s), I guess. These are crucial details, which need to be added to the manuscript.## We have added a connection from the formulas for the linear regressions to measured quantities and then added the equation 1, for the model formula clarity. We believe now the link between the model parameters determination (Figure**

3a) and the modelled spectrogram (Figure 4) is more clear. ## L188 and following: To further stress the point that increased wind speeds result in reduced event detections, I suggest to look at the recorded data and plot the event detections as a function of wind speed. This should show a drop in detections at higher wind speeds, in case the wind doesn't affect the icequake generation processes. Also, the red bars in Fig. 5 may actually be replaced (or compared) by the measured RMS of ELIS, which I assume is not much affected by the few short duration events per 6 hours.## As explained above, the wind speed has different effects on different frequencies. It is therefore important to use the modelled RMS values to compare with the seismic rate. On attached Figure R2, we plot the number of events per 6 h vs the average wind speed per 6h, adding them all together on one plot makes the figure less readable. We also plot (Figure R3) your suggestion of the ELIS RMS vs Wind speed, but we believe the modelled RMS is more appropriate in the manuscript as it demonstrates the usefulness of computing it. The results of Lombardi et al are showing large diurnal RMS variations, that they link with local icequake activity. This activity therefore is largely responsible for the observed RMS, even though each individual icequakes are short duration.

**L199 and following: Here, the wind speed measured at the base station is used to calculate wind-induced noise power, which is subtracted also from the five stations of the temporary network. The temporal variability of the wind-corrected PSDs are then discussed. I think that such an analysis is not well justified, as wind speeds might not be well correlated at the sites (as also indicated in the manuscript). For instance, station ANT6 is separated by about 50 km from the weather station and on the other side of a 4000 m high mountain range, which I expect to clearly influence wind conditions. In addition, the authors find that wind-corrected PSDs are still correlated with wind. This is not surprising considering that the wind-noise model is calculated from the 5th percentile of PSD observations, hence removing not the full contribution.These shortcomings must be discussed in the manuscript. Currently, this is only briefly picked up in the conclusions.## Thank you for this comment. Indeed, the size and context of the ANT array suggest that the wind field might be different overall. We rewrote**

this section and replaced Figure 6 by the non wind-corrected data. We adapted the discussion about potential wind-induced, or icequakes-induced noise sources. The usage of the 5th percentile has the objective to define the baseline of the changes, and in the modelling (no longer used now), we used only the a1 and a2 parameters, so the "changes", and not the baselines (b1 and b2).

**L225 and following: This section discusses the temporal PSD/RMS variations of the wind-corrected stations, but I think that this section does not have a very profound basis given the issues raised in the previous point. Also, I doubt the usefulness of analyzing RMS amplitudes in the context of discrete icequake events. The events presented in Lombardi et al. 2019 are of short duration (<1s) and weak amplitude (<1e-6m/s), hence, I do not expect them to cause a significant contribution to the RMS amplitude. Yet, this could be checked by running a simple STA/LTA trigger on e.g. station ANT6 during high RMS amplitude periods, which is argued to register more seismicity due to its deployment on blue ice. Overall, I feel that the discussion of RMS amplitudes, does not yield useful insights into glaciologically relevant processes. Given the uncertainties and potential overinterpretation, I suggest to significantly shorten the discussion of RMS variations in the light of ice flow dynamics. Instead, I suggest to discuss some other aspects as detailed in the following comment.## The results of Lombardi et al are showing large diurnal RMS variations, that they link with local icequake activity. This activity therefore is largely responsible for the observed RMS, even though each individual icequakes are short duration. We believe our observation of the different patterns of icequakes activity deduced from the RMS amplitudes is relevant to the study of the icequakes in the area. If the icequakes would be very shallow events due to the thermal expansion of the ice, then indeed those are probably not relevant for, e.g., basal processes, but they are still witnesses of the state of the ice. ## Wind-induced noise levels: I am missing a discussion of the wind-noise levels and comparison to other studies. For instance, how do the results compare to the findings of the cited study by Lott et al. (2017), who also analyze wind-induced noise as a function of wind speed? In this context, it would also be helpful to discuss the wider applicability of**

the derived noise level. Is it also applicable to other sites in Antarctica? If available, it would be also very interesting to study other colocated seismic and weather stations.## Thank you for this comment. We now included a comparison of our wind-noise results to the Lott et al., (2017) wind-noise study. The comparison with other stations/bases would be interesting in future studies. We believe our methodology is simple and can be easily reproduced elsewhere. We also now added the importance of the fact that wind doesn't seem to significantly affect the frequencies between 0.1 to 1.0 Hz, often used for ambient seismic noise-based imaging and monitoring. ## ELIB vs ELIS: Sometimes, the manuscript refers to station ELIS, sometimes to ELIB. According to Table 1, these are two different stations, with ELIB being a borehole station, yet, the existence of such a station is not mentioned in the text. This issue needs to be clarified. Actually, it would be very interesting to see Figure 2 for both surface and borehole station to evaluate the effect of a shallow borehole (according to Table 1, ELIB sits in a depth of roughly 10m?) on the wind-noise level.## Yes, in fact we made mistakes using ELIB instead of ELIS. At PEAS, there were 2 stations : ELIB (borehole) was the first installed in February 2010 followed by ELIS (surface) in February 2012, due to technical difficulties ELIB use was discontinued, therefore the data used in our study was from ELIS and not ELIB. I corrected my mistakes and replaced ELIB by ELIS where it was wrong. Also, it would be very interesting to see the difference between the borehole and the surface seismometer, it could be done but for Figure 2, we used a complete year of seismic records (2017) that was recorded by ELIS but ELIB.

**Corrections to the Minor comments:## L37: I corrected accordingly and replaced PE by PEAS in the whole manuscript (Princess Elisabeth Antarctica Station) to avoid confusion. L38: I rephrased and added an explanation of why Antarctica is extensively used in meteorite finding: "The PEAS allowed investigation in the field of meteorites as spotting them on the emptiness of Antarctica is simpler than mixed up with vegetation and rocks". L46: This station increases the sparse coverage of seismic stations in the Sør Rondane mountain range in Antarctica L64-65: At PEAS, a few anthropogenic noise sources exist year-round like wind turbines, and seasonal human activities outside and inside the buildings during the summer L99: Yes, I failed to mention that the first instrument installed in 2010 was installed in a 13 m borehole and due to failing of the instrument and the high-maintenance cost, it was then replaced in February 2012 by a broadband seismometer at the surface. L102: It is continuous, I wrote partly because the data was continuous for 2019 and 2020 but had yet to be downloaded from the station. L118-119: Yes, you are right, I use Obspy and made a mistake describing it. I corrected it. L138: I didn't write it correctly. By maximum average, I mean that the Weather Station records and the maximum wind speed recorded every 10 minutes which is averaged over an hour. I rewrote it to "the average of the maximum" L138: 25 m/s represents a threshold whereas over the period studied, the wind-speed didn't exceed it. L145: Yes, I made a mistake by saying ground velocity, I corrected it to ground acceleration. L160: Explain the Perseval's theorem? L164: It is also the RMS amplitude used in Lombardi et al., 2019 L196-198: The new sentence is as follows : "If these events were equally distributed over the time-period, ∼7.9 events could be detected each day but, the numbers of events found over this period were not, as for example, during the 3rd of February 5 events were detected whereas, on the 4th of February, not a single event was recorded dur to a drastic increase of the RMS." L199: Yes, in fact, I meant that some of the temporary stations, i.e.: ANT3 observes greater icequakes due to its location near higher ice-flow speed channels. I shortened the sentence to avoid confusion. L258: Yes, I made a mistake with the units explained. I corrected it.**

**Answers to the General comments & Figure comments:## I modified in the whole manuscript "seismic velocity" to "ground velocity" and "thermal icequakes" by "thermally-induced icequakes".**

Table 1 and 2: I merged Table 1 and 2. Figure 4: corrected the citation on the figure and modified the caption accordingly. Figure 3: I modified and expanded the Figure 3 to make it more readable. Figure 5: I changed the red y-axis label from "RMS" to "Modeled RMS". Figure 6: I added labels (a) => (n) and changed the caption to better

explain the graph.

[Figure]

**Fig. 1.** Figure_R1

[Figure]

**Fig. 2.** Figure_R2

[Figure]

**Fig. 3.** Figure_R3

---

## Author Response (AR2)

**Wind-induced seismic noise at the Princess Elisabeth Antarctica Station**

**Point-by-point answer**

Dear Referee 1,

Thank you for your review, which helped us improve the manuscript.

Please find answers to your questions here below.

Baptiste Frankinet,

*For the authors.*
* * *
**Report #1**

Thank you for your comments and suggestions.

We followed your suggestions, in the PDF and in the short list you attached. For the remaining questions, please see hereafter:

**Question 1) Model for wind-induced seismic noise**
In the revised version, you introduced a piecewise continuous function (eq. 1) to describe wind-induced noise. Is the function overall continuous, i.e. also at 6m/s wind speed? From the text, I conclude that the parameters for both function pieces are determined independently from each other via linear regression, in which case the two slopes will very likely intersect at a different wind speed than 6m/s (hence introducing an offset at 6m/s). However, from Fig. 3b, I conclude that the resulting function (red line) is continuous at 6m/s – is there some condition on the continuity of the function, i.e. y1(6m/s)=y2(6m/s)?
Please clarify these issues. ##

Thank you for this remark, the 2 parameters for both functions are determined independently and no continuity was imposed, other than including the 6 m/s value in both linear regressions. As you suggested, we computed the difference between y1 and y2 for a wind velocity of 6 m/s, and then plotted their difference for all periods on the Figure R1 below. As it can be seen the difference is minimal, in the order of 0.01 dB, and the 2 linear functions can be considered continuous.

[Figure]

*Figure 1: R1, the Y axis represents the difference between the noise amplitude predicted by the two linear relations, for a wind speed of 6 m/s, and is expressed in dB.*

**In addition, you state that the model is frequency dependent, yet, in the abstract you present values for the slopes in dB/m/s without any frequency context.**

We adapted the abstract, the period was lacking indeed, thanks!

**Also, I think it would be good to briefly state that the seismic power appears to scale to first order linearly with wind speed or something similar. In the current version, it sounds a bit that it would be a given that there is a linear relationship.**
**Finally, it is not clear to me, why the RMS amplitude of ground velocity calculated from the model increases exponentially with wind noise (Fig. 4b), whereas the acceleration PSD scales linearly with wind noise. Where does the exponential relationship come from?**

The relation between wind speed and seismic *power* is linear, therefore the relation is exponential between wind speed and seismic *amplitude*. We added a sentence in the abstract too, to make sure those linear/power relationships are properly understood.

**2. Presentation quality**
**Most importantly, the spectrograms and fonts of Fig. 6 are way too small and make the figure unreadable.**

Thank you for this comment, indeed the fonts were still too small and to improve the readability of Figure 6, we removed the RMS for all stations except ELIS. The figure now focusses on the main information visible inside the spectrograms, which are now large enough.

**In addition, the quality of the manuscript's text would greatly benefit from a thorough proofread. Many parts are written imprecisely and suffer from grammatical errors. Some sentences are hardly understandable, e.g. „The wind speed used in this study is the average of the maximum wind speed recorded by the AWS which is measured at 2 m height every 10 minutes and averaged for each hour of 2017." Please revise the text. ##**

We also did a proofread of the manuscript and improved the discussion part were some sentences needed to be improved.

**## 3. I still think it might be worth to show the icequake detection rate as a function of the wind speed. In their response letter, the authors present a similar figure, but both quantities as a function of time. To stress the point that less icequakes are detected at high wind speeds, it would be better to plot the icequake rate (y-axis) directly as a function of the wind speed (x-axis), independent of time. This would stress the point that wind puts a bias on icequake detections and would strengthen the results of the paper. ##**

Indeed, thank you for this suggestion to add information about the relation between the seismic rate and the average wind speed, independent of time. We therefore added a Figure 5 b and a description showing the relation between the icequake rate (y-axis) and the wind speed (x-axis). This figure can be seen below (R2) and is added as Figure 5 b) in the final manuscript.

[Figure]

*Figure 2 : R2*

**Other comments**

**- Introduction: give reference for the crevasse propagation velocity**

I have added reference

**- Sentences like „owned by ETH-Zurich" and „map was drawn with QGIS" should probably be moved to the acknowledgements section**

I moved these sentences to the acknowledgements.

**- The formatting of equation 1 and its explanation need to be reworked e.g. say „… where x is the wind speed ..." instead of „x = wind speed"**

I reviewed the Equation 1 formatting and its explanation below.

**- In connection with Fig. 5, „cumulative" number of icequakes is wrong, as it is actually an icequake rate**

This is correct, I modified the text accordingly.

**- „472 events were manually detected by Thierry Camelbeeck (Lombardi et al., 2019)."**
**It is not relevant here, who detected the events, delete!**

I removed the irrelevant information.

**- Fig3, caption: the last sentence does not make sense**

I modified the Fig3 caption to make it clearer.

**- I think the term spectrogram is not correct. You are calculating noise levels as a function of frequency and wind speed, but that's not really a spectrogram (nor a seismogram, which is also mentioned once in the text, I believe).**

This is true, I replace « synthetic spectrogram » by « synthetic frequency and wind-speed dependent noise model »

**- Table 1: the elevation of the automatic weather station is missing**

This was indeed missing, I therefore added the elevation of the automatic weather station to the Table1

I hope these modifications will have improved the quality of our manuscript.

Kind regards,

Baptiste Frankinet

---

## Author Response (AR3)

**Wind-induced seismic noise at the Princess Elisabeth Antarctica Station**

**Point-by-point answer**

Dear Referee,

Thank you again for your review, which helped us improve the manuscript.

Baptiste Frankinet,

*For the authors.*
* * *
**Report #1**

Thank you for your comments and suggestions.

**Technical corrections :**

**In abstract, second last sentence: "model" is mentioned twice, remove the first one.**

Yes, in fact it was repetitive so, we modified accordingly.

**Table 1: The elevation of the AWS has a minus sign, which I believe should not be there.**

There was no minus sign for the AWS elevation.

**a and b parameter of the model: I think there are some formatting issues (e.g. "1" as subscript and after that also a "1" as normal character?!) which should be resolved (fine to do this during the typesetting process).**

We modified the figure 3 accordingly to have only subscript as in the equation and the text.

I hope these modifications will have improved the quality of our manuscript.

Thank you again for your review.

Kind regards,

Baptiste Frankinet